# Condition Monitoring with Prediction Based on Diesel Engine Oil Analysis: A Case Study for Urban Buses

**Hugo Raposo** [1,2,*] **, José Torres Farinha** [1,2] **, Inácio Fonseca** [1,2] **and L. Andrade Ferreira** [3]

1   Centre for Mechanical Engineering, Materials and Processes (CEMMPRE), University of Coimbra, 3030-788 Coimbra, Portugal; torres.farinha@dem.uc.pt (J.T.F.); inaciosaf@gmail.com (I.F.)
2   Instituto Superior de Engenharia de Coimbra, Polytechnic of Coimbra, 3030-199 Coimbra, Portugal
3   Faculty of Engineering, University of Porto, 4200-465 Porto, Portugal; lferreir@fe.up.pt
*   Correspondence: hugrap@gmail.com

**Abstract:** This paper presents a case study and a model to predict maintenance interventions based on condition monitoring of diesel engine oil in urban buses by accompanying the evolution of its degradation. Many times, under normal functioning conditions, the properties of the lubricants, based on the intervals that manufacturers recommend for its change, are within normal and safety conditions. Then, if the lubricants' oil condition is adequately accompanied, until reaching the degradation limits, the intervals of oil replacement can be enlarged, meaning that the buses' availability increases, as well as their corresponding production time. Based on this assumption, a mathematical model to follow and to manage the oil condition is presented, in order to predict the next intervention with the maximum time between them, which means the maximum availability.

**Keywords:** condition monitoring; predictive maintenance; oil analysis; urban buses

## 1. Introduction

Public transportation, in general, and city bus passenger transportation, in particular, represents an important alternative to the use of individual transportation. For this reason, it is essential to focus on the quality of service provided by the public transportation network in order to make it attractive for their users.

Currently, public transportation users are increasingly demanding about the quality of service, so the maintenance stands out as a competitiveness key factor.

Condition monitoring maintenance is the maintenance carried out by using an evaluation of the equipment state, that appeared in the 70s and 80s to designate a new approach to planned maintenance based on condition monitoring techniques [1,2].

According to Pinto [3], the implementation of a condition monitoring system requires investment in equipment, specialized human resources, and specific knowledge. These systems are supported by computer tools that enable, in an efficient way, the analysis, study, recording and control of the data obtained, and also the establishment of some fault trend curves.

In condition monitoring, a common practice is based on recording the equipment condition, reading data in regular intervals and, when the data reading is greater than a value previously defined, the physical asset supervised is considered in fault and a Working Order is launched. In spite of that, they have not been paying enough attention to the relation between that critical interval and their costs reduction potential [4].

Maintenance, in general, and the condition monitoring, in particular, aims to combine increase of reliability with the lowest costs possible, being direct or indirect. In this type of maintenance,

the ecological variables may overlap with the remaining ones. However, conventional indicators are not always fully compatible with environmental indicators [5].

According to Ferreira, increasing availability implies reducing the number of breakdowns, repairs, and inspection times: he adds that it is not enough to have reliable equipment to obtain high availability rates. It is also necessary to ensure maximum speed in repairs, maintenance, and inspection operations [6].

Ahmad and Kamaruddin say that the most emphasized aspect of condition monitoring is the deterioration behavior of the assets. Despite the importance of this aspect, the monitoring of the quality of the maintenance decision process is strategic [7].

An integrating approach of the Life Cycle of a physical asset can be seen in Farinha [8], including management standards like ISO 55000X, interconnected with some econometric models to evaluate their Life Cycle Costs [8].

There are several approaches to forecast the evolution of oil degradation. Newell discusses an approach based on trend analysis to maximize oil change intervals. This author considers the following common oil analysis tests and procedures: viscosity; Total Acid Number (TAN); Total Base Number (TBN); water content; specific gravity; particle count (visual method); spectrometric analysis; ferrographic analysis, [9]. Macián et al. [10] present an analytical approach aiming at more accurate wear determination from engine oil samples. The authors ask: "What level of wear rate is normal or abnormal for the engine studied?" To answer the question, they propose a comparative parameter Z. It considers the engines' deviations between the values of the wear rate and the reference rate. They have also made similar comparisons for a group of engines of the same model. Finally, the influence of the oil consumption and the concentration of the contaminants is linked [10].

Vališ, Zák, and Pokora [11] concentrate on metal particles, such as iron (Fe) and lead (Pb), as potential failure indicators. They apply a linear regression model to determine a linear course of Fe and Pb particle generation. They assume a stochastic process with time dependence. The authors finish with the importance of time series comparisons: Auto Regression Integrating Moving Average (ARIMA); Auto Regression Moving Average (ARMA) methods [11]. Changsong et al. [12] present a study based on 50 oil samples collected and analyzed in sequence covering 250 motor hours. The results show that maintenance intervals can be longer and, at the same time, the cost-effectiveness maintenance ratio can be increased [12].

According to Macián et al. [13], Low Viscosity Oils (LVO) are very important to reduce the fuel consumption in Internal Combustion Engines (ICE). The use of LVO may imply a different tribology behaviour of the engines. The authors tested 39 buses, two different technologies, and four dissimilar lubricants [13].

Macián et al. [14] present a case study of urban buses aiming to evaluate the effect of Low Viscosity Oils (LVO) on fuel consumption and $CO_2$ emissions. They used 39 buses that ran 60,000 km during which they changed the engine oil twice. For 9 of those buses on the second round of oil changing, the effects on oil in fuel consumption and the engine performance were evaluated. The results showed that the LVO reduces the fuel consumption and the $CO_2$ emissions. However, the author says that the engines must be accompanied carefully with high levels of working stress [14].

Macián et al. [15] show that LVO performs well and that oil deterioration depends on the engine technology. In the case of Compressed Natural Gas engines, it was observed that the oil degradation increases [15].

Tormos et al. [16] say that the environmental concerns imply the improvement of the engines' technology. The authors present a tribology model tested in laboratory. Additionally, they also present the potential of the predictive maintenance model [16].

Nowadays, in fleet vehicles, a certain percentage of biodiesel is used. Several authors are not in agreement with the influence of this new type of fuel on oil degradation and, by consequence, on the maintenance based on condition monitoring. Some examples of papers that analyze the effect of these kinds of fuels on oil are reported in [17–21].

Some relevant topics about related works on condition monitoring and predictive maintenance based on oil analysis, in urban buses fleets, are mentioned and discussed in [22–26].

Multivariate statistics are also useful because oil analysis involves several variables. An approach like this may help with the diagnosis of the health of diesel engines. Additional references are referred in [27–29].

This paper presents an approach to the analysis of lubricating oils for Diesel engines, through mathematical and statistical models, namely: exponential smoothing; t-Student distribution; and hypotheses tests validated on some models of urban buses, as it is described throughout the paper.

The paper starts with a global analysis on the importance of oil analysis in predictive maintenance based on condition monitoring. After that, examples of spreadsheets of oil analysis, with emphasis on some important variables, are presented.

Based on the preceding approach, the paper follows the next steps:

- First, the mathematical model to help predicting the next intervention based on Exponential Smoothing, the variable Fe is summarily presented as example;
- Due to the variation of the variable Fe, a t-Student distribution with bilateral test of hypotheses is used;
- Then an example using several values for smoothing parameter and some levels of significance is presented;
- Finally, the influence of the maintenance policy, namely the predictive in the reserve fleet is discussed.

## 2. Condition Monitoring with Prediction through Oil Analysis

The main physical feature of lubricating oil is viscosity and its variation with temperature, given by the viscosity index and the density.

In recent years, there has been a demand for high-performance engine lubricants, especially in the aerospace and automotive industries. This has led to the development of synthetic lubricants which can be maintained at high temperatures without decomposition and have a low risk of combustion.

The synthetic oils are produced using highly refined processes and sophisticated formulations. They derive from synthetic compounds based on PAO (poly-alpha-olefin), non-synthetic PAO, esters, alkylated naphthalene, and alkylated benzene. The use of synthetic oils has become more important in areas where the use of mineral oils does not meet the required needs.

Lubricating oils can cause serious environmental problems if they are discharged indiscriminately, polluting rivers and groundwater. The improper burning of oil adds oxides and toxic gases to the atmosphere. Accordingly, the manufacturers of additives and lubricating oils have been developing products with a longer lifespan, as this tends to reduce oil discharges along the equipment's life cycle.

A key feature of lubricants is their behaviour with increasing temperature: the temperature and pressure are often high. The oils undergo a change when the temperature increases, and their degradation under operating conditions is a problem involving significant economic losses. To report certain special properties of the oil, or to improve the existing ones, especially when the lubricant is subjected to severe working conditions, chemicals are added (additives). The degradation of a lubricant is not an instantaneous process—the loss of its physicochemical properties and contamination are progressive over time and with the use of equipment along its lifetime. Lubricant degradation is affected by the following: oxidation; viscosity variation; contamination; loss of additives (anti-corrosion, anti-wear, dispersing agents, etc.) [30,31].

Today's high-performance lubricants do more than simply reduce friction and wear: they control the generation of deposits, control airborne contaminants, protect against corrosion, have a cleaning function, and maintain the proper operating temperature [30,31].

*2.1. Oil Analysis*

Under certain conditions, a lubricant can deteriorate and no longer fulfil its intended function. It is generally a function concerning the length of the service, the temperature of the system, the environmental conditions, or the stress that it is suffering from, which can often be traced to the presence of dirt or water, acidity, insufficient flow, or inappropriate levels of viscosity. Any of these can cause lubricated components to malfunction. Even when the lubrication system is well designed and maintained, breakdowns can occur in the component, resulting in the deterioration of the lubricant. The deterioration can be chemical or physical, generated internally by the lubricant or by external phenomena. Physical deterioration, often called contamination, materializes as foreign matter in the lubricant, such as water, foundry sand, weld slag particles, metal shavings, dust, and abrasive wear particles.

Lubricant analysis is regularly performed in some industries [31]. It involves four basic steps:

1. Obtaining a sample

Collection of a representative sample of a lubricant, observing certain precautions such as using clean and dry containers; taking extreme care during collection to prevent external contamination; taking samples at operating temperatures [30,31].

2. Identifying a relevant sample.
3. Performing physical-chemical analysis:

Degree of physical and chemical deterioration, i.e., the degree of contamination and degradation, can be evaluated using a set of standard and specialized tests, such as measuring certain properties and comparing these with a baseline value.

Analysis can measure several properties of the lubricant and evaluate their degradation. These include antifreeze; appearance; fuel; content water; soot; nitration; oxidation; sulfation; viscosity; viscosity index; Total Base Number; wear metals (Al content, Cr, Fe, Mo, Na, Ni, Pb, Si, Sn, V); particles.

Particle Quantification (PQ Index) is the measurement of total ferrous (Iron) particles present in the sample. PQ does not take into account the size of particles. The ferrous is detected via magnetic fields and is dependent on the type of laboratory equipment used. They will determine how the measurement is taken. Regardless of this, the generated reading will report the total concentration of the magnetic particles in that sample.

PQ Index can be used to measure ferrous wear metal particles in oil, grease, and coolants. PQ analyzers have no units and can be thought of as mass ferrous particles per mass of oil (Mass/Volume). PQ does not take into account particle size; we need to use the iron (Fe) readings of the elemental analysis to figure out when the concentration level is above 10 μm. This is where the PQ information can be very useful, especially in components that are starting to show fatigue signals or have large internal wear indicators starting to appear rapidly.

4. Interpreting results—diagnosis.
5. Validating diagnosis:

Frequency in which each lubricant ought to be checked depends on various operational factors such as: importance of the equipment; total time of service; scale of production; security; time until failure after detection.

In this section the analysis of service lubricants is addressed, by monitoring the evolution of the degradation of the Diesel oils in the bus fleet, that will have three well-defined phases:

1. In the first phase, the vehicles targeted for analysis and monitoring in the evolution of the degradation of the oils will be selected—this monitoring will be done through the periodic collection of oil samples of the selected vehicles and they will be sent to a proper place for their analysis;

2. In the second phase, an in-depth study of the results obtained in the analyses, as well as of the prediction algorithms to be used, will be carried out;
3. In the third and final phase, an analysis of planned maintenance aiming its improvement will be made. This shall take into account the results obtained in monitoring the evolution of oil degradation—this phase will also serve to present proposals for the improvement of planned maintenance schedules and the reduction of costs that come from it.

This monitoring was done through periodic collection of oil samples from the various vehicles selected and, since there was a small number of samples collected during the period in which this monitoring was developed, the need to use data from older samples belonging to the same homogeneous group was felt. In this research, 10 standard (12 m) urban passenger transport vehicles of three different brands were studied, having been analyzed and studied, 60 oil samples.

These samples helped supporting the studies carried out and proved the relevance of the oil analysis to predictive maintenance based on oil condition monitoring as well as to support new maintenance planning to be used by the company in the future.

These samples were sent to a laboratory located in a European country, with all the characteristics of the vehicle and the oil, such as:

- Number of the vehicle;
- Brand;
- Model;
- Type of car;
- Organ—Motor;
- Equipment km;
- km of the oil;
- Sample date;
- Date of submission of the sample.

Subsequently, reports of the results obtained from the various analyses carried out by the samples collected were received (Figure 1)—this is the original document with the results sent by the laboratory.

These analysis reports allow them to control various properties of the lubricants and to evaluate their degradation throughout the life cycle of the equipment. From them, they can also follow the history of the analysis carried out over time. These include: antifreeze; appearance; fuel; content water; soot; nitration; oxidation; sulfation; viscosity; viscosity index; Total Base Number (TBN); wear metals (Al content, Cr, Fe, Mo, Na, Ni, Pb, Si, Sn, V); particles. Figure 1 illustrates the history of the diverse variables studied.

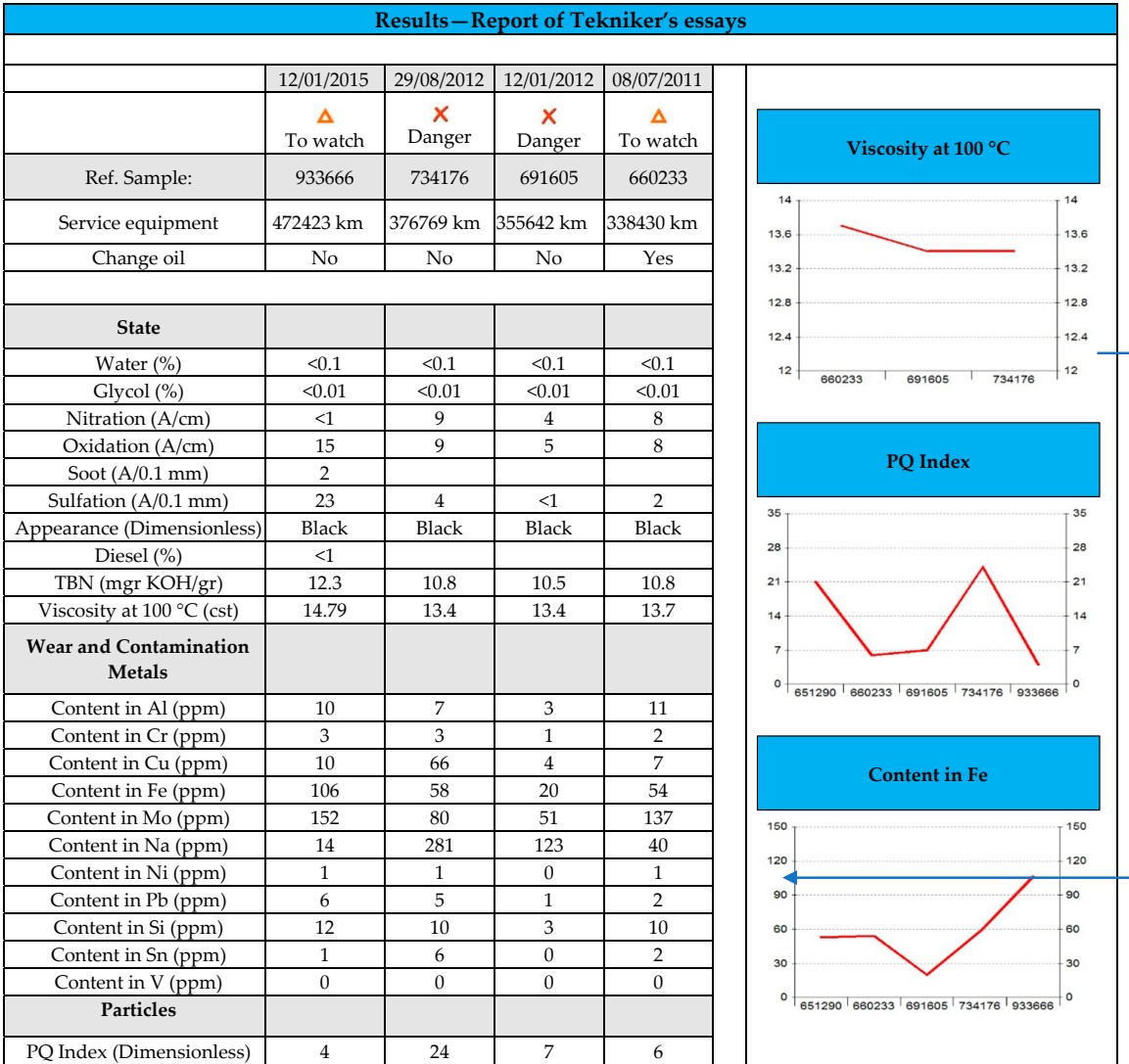

| Results—Report of Tekniker's essays | | | | | |
|---|---|---|---|---|---|
| | 12/01/2015 | 29/08/2012 | 12/01/2012 | 08/07/2011 | |
| | ⚠ To watch | ✗ Danger | ✗ Danger | ⚠ To watch | |
| Ref. Sample: | 933666 | 734176 | 691605 | 660233 | |
| Service equipment | 472423 km | 376769 km | 355642 km | 338430 km | |
| Change oil | No | No | No | Yes | |
| | | | | | |
| **State** | | | | | |
| Water (%) | <0.1 | <0.1 | <0.1 | <0.1 | |
| Glycol (%) | <0.01 | <0.01 | <0.01 | <0.01 | |
| Nitration (A/cm) | <1 | 9 | 4 | 8 | |
| Oxidation (A/cm) | 15 | 9 | 5 | 8 | |
| Soot (A/0.1 mm) | 2 | | | | |
| Sulfation (A/0.1 mm) | 23 | 4 | <1 | 2 | |
| Appearance (Dimensionless) | Black | Black | Black | Black | |
| Diesel (%) | <1 | | | | |
| TBN (mgr KOH/gr) | 12.3 | 10.8 | 10.5 | 10.8 | |
| Viscosity at 100 °C (cst) | 14.79 | 13.4 | 13.4 | 13.7 | |
| **Wear and Contamination Metals** | | | | | |
| Content in Al (ppm) | 10 | 7 | 3 | 11 | |
| Content in Cr (ppm) | 3 | 3 | 1 | 2 | |
| Content in Cu (ppm) | 10 | 66 | 4 | 7 | |
| Content in Fe (ppm) | 106 | 58 | 20 | 54 | |
| Content in Mo (ppm) | 152 | 80 | 51 | 137 | |
| Content in Na (ppm) | 14 | 281 | 123 | 40 | |
| Content in Ni (ppm) | 1 | 1 | 0 | 1 | |
| Content in Pb (ppm) | 6 | 5 | 1 | 2 | |
| Content in Si (ppm) | 12 | 10 | 3 | 10 | |
| Content in Sn (ppm) | 1 | 6 | 0 | 2 | |
| Content in V (ppm) | 0 | 0 | 0 | 0 | |
| **Particles** | | | | | |
| PQ Index (Dimensionless) | 4 | 24 | 7 | 6 | |

**Figure 1.** Oil analysis results report-Bus.

The data collected was entered into an Excel spreadsheet, in order to create a database where they could be easily analysed.

Figure 2 shows an Excel database example, per vehicle. In this figure, the historical data of the collected analysis referring to the equipment can be verified, as well as the identification and the characteristics of the vehicle studied.

| Lubricants Analysis | | | | | |
|---|---|---|---|---|---|
| **Number Fleet** 287 | | | | | |
| | | Equipment Data | | | |
| Registration: 64-30-XG | | VOLVO | | **Model:** | B 7 L |
| | | Lubricant Data | | | |
| | | | | √ | Normal |
| Lubricant: | XXX | | | Δ | To watch |
| | | | | X | Danger |
| | | Result of Samples | | | |

| Date | | 27/07/2007 | 10/10/2007 | 02/01/2008 | 14/02/2008 | 14/04/2008 |
|---|---|---|---|---|---|---|
| Reference sample | | 373108 | 383767 | 396906 | 403855 | 414581 |
| km do Equipment | | 102 259 | 175 819 | 186 535 | 192 571 | 198 046 |
| km do Lubricant | | 15 000 | 15 000 | 10 000 | 15 000 | 20 000 |
| | | | | | | |
| **State** | | | | | | |
| Antifreeze (%) | (PE-TA.071) | 0.08 | 0.08 | 0.08 | 0.08 | 0.08 |
| Appearance (adim) | (PE-TA.096) | Black | Black | Black | Black | Black |
| Fuel (%) | (PE-TA.071) | 2 | 2 | 2 | 2 | 2 |
| Water content (%) | (PE-TA.071) | 0.1 | 0.1 | 0.1 | 0.1 | 0.1 |
| Water content (FinachecK) (%) | (PE-5022-Al) | | | | | |
| Soot (%) | (DIN 51452) | 2 | 2.9 | 2.3 | 2.5 | 2.5 |
| Nitration (ABS / cm) | (PE-TA.071) | 6 | 4 | 5 | 6 | 8 |
| Oxidation (ABS / cm) | (PE-TA.071) | 2 | 5 | 2 | 2 | 1 |
| Sulfation (ABS / cm) | (PE-TA.071) | 6 | 1 | 6 | 6 | 8 |
| TBN (mgr KOH / g) | (ASTM D-2896-07a) | 10.2 | 10.52 | 10.86 | 10.2 | 8.9 |
| Viscosity at 100 ° C (cst) | (ASTM D-445-11) | 13.4 | 14 | 13.3 | 13.4 | 13.7 |
| | | | | | | |
| **Wear and Contamination Metals** | | | | | | |
| Content in Al (ppm) | (ASTM D-5185-05 mod.) | 2 | 3 | 3 | 2 | 2 |
| Content in Cr (ppm) | (ASTM D-5185-05 mod.) | 1 | 1 | 1 | 1 | 1 |
| Content in Cu (ppm) | (ASTM D-5185-05 mod.) | 1 | 2 | 2 | 1 | 3 |
| Content in Fe (ppm) | (ASTM D-5185-05 mod.) | 24 | 35 | 28 | 31 | 56 |
| Content in Mo (ppm) | (ASTM D-5185-05 mod.) | 2 | 2 | 3 | 2 | 2 |
| Content in Na (ppm) | (ASTM D-5185-05 mod.) | 0 | 9 | 14 | 3 | 9 |
| Content in Ni (ppm) | (ASTM D-5185-05 mod.) | 0 | 0 | 1 | 0 | 1 |
| Content in Pb (ppm) | (ASTM D-5185-05 mod.) | 4 | 2 | 1 | 3 | 3 |
| Content in Si (ppm) | (ASTM D-5185-05 mod.) | 14 | 12 | 10 | 6 | 10 |
| Content in Sn (ppm) | (ASTM D-5185-05 mod.) | 0 | 0 | 0 | 0 | 1 |
| Content in V (ppm) | (ASTM D-5185-05 mod.) | 0 | 0 | 0 | 0 | 0 |
| | | | | | | |
| **Particles** | | | | | | |
| PQ Index (Adim) | (PE-5024-Al) | 25 | 9 | 16 | 6 | 16 |
| | | | | | | |
| **Diagnosis** | | | | | | |
| Sample Diagnosis | | Δ | Δ | X | Δ | Δ |

**Figure 2.** Database in Excel.

The data entry is no more than the various results obtained when analyzing the variables that characterize the lubricants. In this phase, the variables were analyzed using the method presented in Section 4, which allows monitoring the evolution of oils degradation.

All variables were studied. However, this paper will only focus on those which were considered the most important for the monitoring of the degradation of oils, which are:

- Soot (Carbon Matter);
- Viscosity;
- TBN;
- Wear and Contamination Metals;
- Particles.

Therefore, for the study of the variables used as reference, the limits made available by the laboratory were used, as can be seen in Table 1.

**Table 1.** Limits for the various parameters.

| Characteristics of the Oil | | Limits (X > Danger) |
|---|---|---|
| Antifreeze (%) | (PE-TA.071) | 0.08 |
| Appearance (dimensionless) | (PE-TA.096) | |
| Fuel (%) | (PE-TA.071) | 4.0 |
| Water content (%) | (PE-TA.071) | 0.2 |
| Water content (FinachecK) (%) | (PE-5022-Al) | 0.2 |
| Soot (%) | (DIN 51452) | 1.5 |
| Nitration (ABS/cm) | (PE-TA.071) | 15 |
| Oxidation (ABS/cm) | (PE-TA.071) | 15 |
| Sulfation (ABS/cm) | (PE-TA.071) | 20 |
| TBN (mgr KOH/g) | (ASTM D-2896-07a) | 30 |
| Viscosity at 100 °C (cst) | (ASTM D-445-11) | 15 |
| **Wear and Contamination Metals** | | **Limits** |
| Content in Al (ppm) | (ASTM D-5185-05 mod.) | 20 |
| Content in Cr (ppm) | (ASTM D-5185-05 mod.) | 10 |
| Content in Cu (ppm) | (ASTM D-5185-05 mod.) | 35 |
| Content in Fe (ppm) | (ASTM D-5185-05 mod.) | 90 |
| Content in Mo (ppm) | (ASTM D-5185-05 mod.) | 20 |
| Content in Na (ppm) | (ASTM D-5185-05 mod.) | 40 |
| Content in Ni (ppm) | (ASTM D-5185-05 mod.) | 20 |
| Content in Pb (ppm) | (ASTM D-5185-05 mod.) | 40 |
| Content in Si (ppm) | (ASTM D-5185-05 mod.) | 20 |
| Content in Sn (ppm) | (ASTM D-5185-05 mod.) | 15 |
| Content in V (ppm) | (ASTM D-5185-05 mod.) | 00 |
| **Particles** | | **Limits** |
| PQ Index (Dimensionless) | (PE-5024-Al) | 110 |

The Iron content (ppm) was considered one of the most important variables and, because of that, one that was thoroughly studied. This allowed us to draw several conclusions on the state of oil degradation and the equipment, which will be described later.

*2.2. Oil Analysis Changes through Prediction*

In the first step, the exponential smoothing method of the iron content (Fe) was applied in order to determine the evolution of its degradation, as can be seen in Table 2 and Figure 3. The table and graph show a clear degradation in the iron content of the analysed oils. Obviously, the prediction of the next values will involve increased degradation. When this variable has values like those shown in the table, the oil must be replaced, because the equipment is at a high risk level.

The main formula for exponential smoothing is given by:

$$S_{t+1} = \beta X_t + (1-\beta)S_t \iff S_{t+1} = \beta \sum_{i=0}^{t} (1-\beta)^i X_{t-i} \tag{1}$$

A smoothing parameter, $\beta$ corresponding to the history of the variable of concern is required to forecast its value in the next period [32].

$$0 \leq \beta \leq 1$$

where: $S_{t+1}$ is the forecast for the next time; $X_t$ is the real value recorded in the present time; $S_t$ is the forecasted value for the present time; $\beta$ is the smoothing parameter.

**Table 2.** Application of exponential smoothing-Fe (ppm) content.

| | Fe Content (ppm) | | | |
|---|---|---|---|---|
| **Period km** | **Observed Value** | **Prediction with $\beta$ = 0.1** | **Prediction with $\beta$ = 0.5** | **Prediction with $\beta$ = 0.9** |
| 2451 | 19 | | | |
| 5214 | 53 | 19.00 | 19.00 | 19.00 |
| 10,115 | 22 | 22.40 | 36.00 | 49.60 |
| 12,403 | 14 | 22.36 | 29.00 | 24.76 |
| 17,212 | 54 | 21.52 | 21.50 | 15.08 |
| 22,183 | 141 | 24.77 | 37.75 | 50.11 |
| 27,682 | 28 | 36.39 | 89.38 | 131.91 |
| 30,965 | 77 | 35.55 | 58.69 | 38.39 |
| 35,965 | | 39.70 | 67.84 | 73.14 |

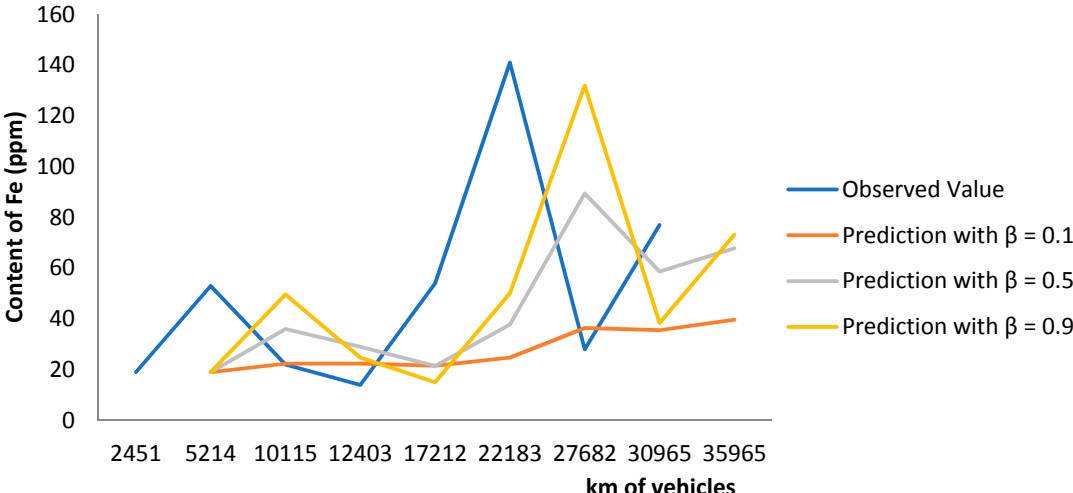

**Figure 3.** Graph of exponential smoothing—Fe (ppm) content.

In the second part of the development of algorithms for the model, the data was analyzed using the t-student distribution, $n \leq 30$, [10].

$$\mu = \overline{X} + t_\alpha * \frac{S}{\sqrt{n}} \tag{2}$$

where,

$\mu$   Is a fixed value used for comparison with the sample mean;

$\overline{X}$   Is the average sample;

$t_\alpha$   Corresponds to the critical T;

*S*　Is the sample standard deviation;

*n*　Is the sample size.

where $t_\alpha$ corresponds to the critical *t* of a tail, considering the desired confidence interval, and the degrees of freedom $n - 1$.

Next, we followed up on the evolution of the degradation of this variable using the t-student distribution. The goal was to estimate the average value of iron (Fe). As Table 3 shows, the average of Fe content was 51 (ppm). This is above the upper normal limits, suggesting a high level of degradation. Note that through additional methods, like the t-student test, it is possible to calculate other important data, such as the sample mean, sample standard deviation, and upper parameter limit for several confidence intervals.

**Table 3.** Application of t-student test to iron content—Fe (ppm).

| | | | Content Fe (ppm) t-Student | | |
|---|---|---|---|---|---|
| | $\alpha = 0.001$ | $\alpha = 0.01$ | $\alpha = 0.05$ | $\alpha = 0.1$ | $\alpha = 0.2$ |
| **Average (sample)** $\overline{X}$ | 51.00 | 51.00 | 51.00 | 51.00 | 51.00 |
| **Standard deviation (sample)** *S* | 42.35 | 42.35 | 42.35 | 42.35 | 42.35 |
| **Critical** *t* | 4.79 | 3.00 | 1.89 | 1.41 | 0.90 |
| **Standard deviation (population)** $\sigma$ | 46.27 | 34.83 | 24.63 | 19.19 | 12.60 |
| **Population Average** ($\mu_0$) | 51 + 46.2 | 51 + 34.8 | 51 + 24.6 | 51 + 19.1 | 51 + 12.6 |
| **Upper limit** | 97.27 | 85.83 | 75.63 | 70.19 | 63.60 |

Lastly, bilateral tests of hypotheses for the value of $\mu$ were used:

$$H_0: \mu = \mu_0; H_1: \mu \neq \mu_0$$

$\mu$ is considered a random variable whose distribution for small samples ($n < 30$) is given by:

$$t = \frac{\overline{X} - \mu_0}{\frac{S}{\sqrt{n}}} \tag{3}$$

In general, $\sigma$ (standard deviation of the population) is unknown. The process is the following:

- A one-tailed test uses one threshold value (associated with the chosen significance level) and rejects the hypothesis $H_0$—where $T > T$ *critical*—when the value of the modulus calculated for the *t* statistic exceeds the critical value.

Finally, the average population of the iron was estimated by the following significance levels: 0.001; 0.01; 0.05; 0.1; 0.2. As Table 4 shows, with a value of 80 (ppm) and a confidence interval of 99%, the hypothesis $H_0$ is not rejected. But with a 90% confidence interval, $H_0$ will be rejected because the value of *t* (1.59) is higher than the value of the confidence interval (1.41). Furthermore, with a confidence interval of 80% (0.90) and a sample average of 51.00, the average value is 37.59.

**Table 4.** Application of t-student test to iron content (Fe).

| | | | Hypothesis Test | | |
|---|---|---|---|---|---|
| $\mu_0$ (Population Average) | Calculated t | Table t $\alpha = 0.001$ | Table t $\alpha = 0.05$ | Table t $\alpha = 0.1$ | Table t $\alpha = 0.2$ |
| 25.00 | 1.23 | 4.79 | 1.89 | 1.41 | 0.90 |
| 35.00 | 0.76 | 4.79 | 1.89 | 1.41 | 0.90 |
| 45.00 | 0.28 | 4.79 | 1.89 | 1.41 | 0.90 |
| 50.00 | 0.05 | 4.79 | 1.89 | 1.41 | 0.90 |
| 65.00 | −0.66 | 4.79 | 1.89 | 1.41 | 0.90 |
| 75.00 | −1.13 | 4.79 | 1.89 | 1.41 | 0.90 |
| 80.00 | −1.59 | 4.79 | 1.89 | 1.41 | 0.90 |
| $\mu_0$ | | 20.64 | 22.64 | 29.82 | 37.59 |

## 3. Discussion

With this condition monitoring model several variables can be evaluated, which can help understanding the evolution of the degradation state of the oils. The models exemplified here were applied in three ways:

1. Individually, to all vehicles (all parameters);
2. Homogeneous groups of different vehicles (all parameters);
3. To the groups of vehicles that use biodiesel as fuel (all parameters).

The exponential smoothing was applied to the iron content (Fe) variable for a bus number $XX_3$ to determine the evolution of its degradation. When this variable has high values, the equipment is at a high risk level, and the oil must be changed. The second model applied to monitor the degradation of the iron content was based on the t-student distribution: it estimates the average of iron content (*Fe*)—the average content is 99.80 (ppm).

It is also possible to calculate more information, such as the sample mean, the sample standard deviation, and the upper limit of the parameter to determine the confidence intervals. If the value of 150 (ppm) is found in the iron content variable with a 99% confidence interval, the hypothesis $H_0$ is not rejected. But, if the confidence level is 90%, $H_0$ is rejected. The value of $t$ (2.35) cannot be greater than the value of the confidence interval (1.53).

If the value of $t$ is used from the t-student table with 80% confidence interval (0.90) and a sample mean of 99.80, a mean value for a population of 70.48 is obtained.

Because of the oil itself and the great influence that it has on the diesel engine's condition, the accompaniment of its degradation permits us to maximize the bus availability itself and the bus fleet in general.

The paper demonstrates that by using condition monitoring maintenance, the intervals of the interventions of which can be increased which can, consequently, increase the bus fleet availability, reducing the maintenance costs. Through this study, it was possible to increase the intervals of some models of buses of this company, having as reference intervals of 20,000 km between oil changes. The new intervals are of 25,000 km between each substitution. With this change, a lot of the maintenance costs were reduced.

Table 5 shows the data concerning the company studied, such as the number of buses that constitutes the fleet, their availability, the need of buses for production, the number of buses under maintenance, and the number of buses that correspond to the reserve fleet, based on a systematic preventive maintenance policy.

Figure 4 (radar map) shows the Availability *versus* Production Requirement (buses necessary to carry out the careers) of the company during a year.

**Table 5.** Availability versus Need for buses—Systematic preventive maintenance.

| Months | Bus Fleet | Availability | Need | Maintenance | Reserve Fleet |
|---|---|---|---|---|---|
| January | 115 | 107 | 90 | 18 | 7 |
| February | 115 | 104 | 90 | 21 | 4 |
| March | 115 | 105 | 90 | 19 | 6 |
| April | 115 | 106 | 90 | 18 | 7 |
| May | 115 | 107 | 90 | 18 | 7 |
| June | 115 | 106 | 90 | 19 | 6 |
| July | 115 | 102 | 90 | 22 | 3 |
| August | 115 | 103 | 90 | 22 | 3 |
| September | 115 | 106 | 90 | 19 | 6 |
| October | 115 | 107 | 90 | 18 | 7 |
| November | 115 | 109 | 90 | 16 | 9 |
| December | 115 | 106 | 90 | 18 | 7 |

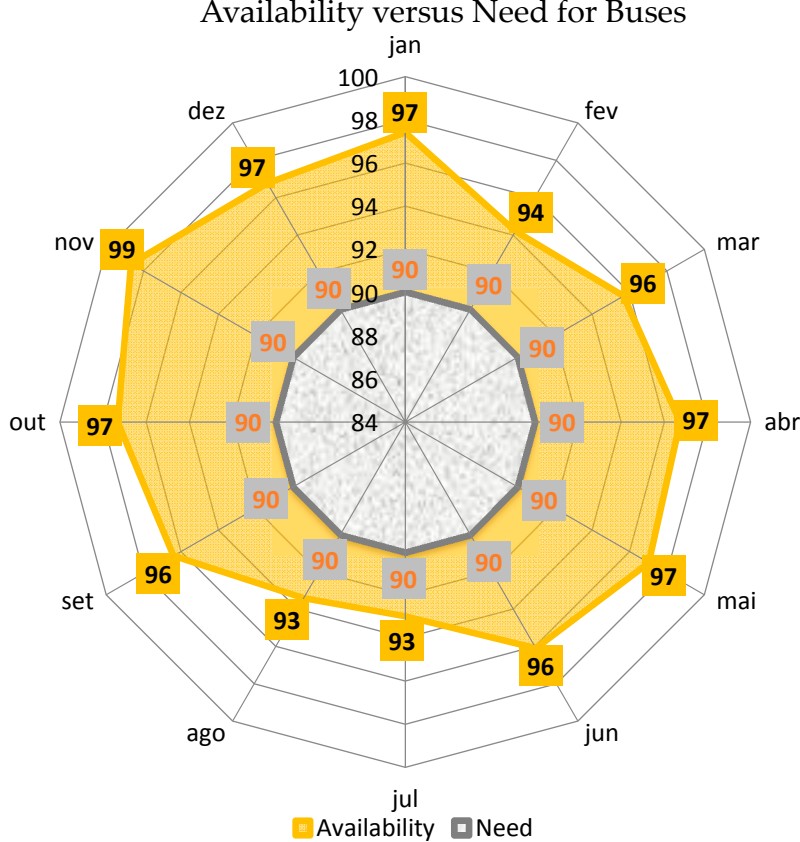

**Figure 4.** Availability versus Need for buses—Systematic Preventive Maintenance.

## 4. Conclusions

This paper shows how the implementation of a condition monitoring based maintenance policy may maximize the physical asset availability, reduce the costs of maintenance, as well as overcharges, and provide an additional guarantee of reliability.

It also shows how some variables, such as soot and iron content, enhance the condition of diesel engines. The analysis can be extended to include other variables.

The paper demonstrates that monitoring oil condition can increase the availability of equipment and improve fault prevention by allowing early intervention in its degradation. It also demonstrates that the implementation of a condition monitoring based maintenance policy, using oil analysis, has huge advantages for public transport companies, including lower downtime, higher availability and, consequently, better service for users.

Additionally, this kind of maintenance policy contributes to the rationalization of the size of a reserve fleet.

Finally, the methodology can be used for many other types of physical assets.

**Author Contributions:** Investigation, H.R., J.T.F., I.F. and L.A.F.

**Funding:** This research received no external funding

**Conflicts of Interest:** The authors declare no conflict of interest.

## Symbols and Acronyms

| | |
|---|---|
| $\beta$ | Is the smoothing parameter |
| $X_t$ | Is the real value recorded in the present time |
| $\mu$ | Is a fixed value used for comparison with the sample mean |
| $\overline{X}$ | Is the average sample |

| | |
|---|---|
| $t_\alpha$ | Corresponds to the critical T |
| n | Is the sample size |
| μo | Population Average |
| Al | Aluminium |
| ARIMA | Auto Regressive Integrated Moving Average |
| ARMA | Auto Regressive Moving Average |
| Cr | Chromium |
| Cu | Cobalt |
| Fe | Iron |
| $H_0$ | Hypothesis 0 |
| $H_1$ | Hypothesis 1 |
| ICE | Internal Combustion Engines |
| LVO | Low Viscosity Oils |
| Mo | Molybdenum |
| Na | Sodium |
| Ni | Nickel |
| PAO | Polyolefin, Polyester, polyglycol |
| Pb | Lead |
| S | Is the sample standard deviation |
| Si | Silicon |
| Sn | Tin |
| $S_t$ | Is the forecasted value for the present time |
| $S_{t+1}$ | Is the forecast for the next time |
| TAN | Total Acid Number |
| TBN | Total Base Number |
| V | Vanadium |
| σ | standard deviation of the population |

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
