# Peer review of "Condition Monitoring with Prediction Based on Diesel Engine Oil Analysis: A Case Study for Urban Buses"

_actuators, doi:10.3390/act8010014_

Round 1

Reviewer 1 Report

This paper presents a work that could be of interest to the reading audience of Actuators. However, the document has a lot of content deficiencies. In addition, it is already published with DOI: https://www.preprints.org/manuscript/201811.0167/v1

The paper aims to study and a model for condition monitoring of Diesel engines’ oil of urban buses, through the accompaniment of the evolution of its degradation, with the objective to implement a predictive maintenance policy. It is not clear that the subject of the document in the scope of the journal.

Some of the comments are the following:

GENERAL

1. Add a list of symbols and acronyms. Please add all symbols and acronyms that the authors use in the equations, tables, and text. And the authors could remove lines: 236 to 240, and 248 to 252.

2. References. Minimize the use of documentation in Portuguese as references (there are 8 references in Portuguese, 36% of the total is too much). Some references are somewhat old (1996, 1997, 1999, and 2000). The authors should look for more current references. In general, the number of references is scarce.

3. Abstract should be focused on the main points of this study. In summary there is no information on the methods developed and the results. Please add.

4. Cite references. Please, check all the document and cite the bibliography correctly (lines: 59, 62, 66, 73, 77, 81, 86, 106, etc.). Examples of errors that authors have, and in some cases, how to solve them.

4.1 Line 59. ‘Farinha (2018)’ change by ‘‘Farinha [8]’

4.2. Line 66. ‘Macián et al.’ (what of them?), change by ‘Macián et al. [10]’

4.3. Line 114. ‘additional references: [17], [18] and [19]’ change by ‘additional references [17-19]’

5. Introduction. The introduction should include oil analysis methods, the influence of used fuel, and the objectives of the document.

6. Figures. The graphics quality of the figures is very bad. Figures without legends, with minuscule letter size, no name on the axes, etc. Please, improve these figures.

SPECIFIC

1. Lines 33 to 35. Please move this sentence at the end of the introduction.

2. Please remove the commas before the bibliographic citation. Example: line 35 'techniques [1]' or line 37 'continuously [2]'.

3. Please, join these two sentences in one: ‘The condition monitoring maintenance appeared in the 70-80’s to designating a new approach to planned maintenance based on the knowledge of the state of equipment, using condition monitoring techniques, [1]. Condition monitoring maintenance is the maintenance carried out by means of an evaluation of the equipment state, usually carried out continuously, [2].’ The information is repetitive and superfluous.

4. Line 112. Please remove: ‘Although not the topic of this paper’.

5. Introduction. The authors must add a paragraph at the end of the introduction section with the objectives that the document intends after carrying out the bibliographic review.

6. Currently in fleet vehicles a certain percentage of biodiesel (or other alternative fuels) is used. It would be interesting for the authors to add articles to the introduction that analyze the effect of these fuels on oil. Some examples (of course the authors can look for other articles):

·       B. Tormos, et al. "Results of an Operating Experience for Urban Buses Fuelled with Biodiesel Blends (B50)" SAE paper, vol. 2009-01-1827, 2009.

·       S. O. Shrake, et al. "A comparative analysis of performance and cost metrics associated with a diesel to biodiesel fleet transition" Energy Policy, vol. 38, pp. 7451-7456, 2010.

·       S. Tica, et al. "Test run of biodiesel in public transport system in Belgrade" Energy Policy, vol. 38, pp. 7014-7020, 2010.

·       R. Fraer, et al. "Operating Experience and Teardown Analysis for Engines Operated on Biodiesel Blends (B20)" SAE paper, vol. 2005-01-3641, 2005.

·       K. Proc, et al. "100,000-Mile Evaluation of Transit Buses Operated on Biodiesel Blends (B20)" SAE paper, vol. 2006-01-3253, 2006.

7. Lines 118-121. Please add the studies that support this premise.

8. Lines 122-125. Please add the studies that support this premise.

9. Lines 126-130. Please add the studies that support this premise.

10. Lines 140-142. Please add the studies that support this premise.

11. It would be interesting if the introduction of section 2 (Lines 116-142) was included in section 1.

12. Line 144-152. Please add the studies that support this paragraph.

13. Line 184. This sentence: ‘This monitoring was done through a periodic collection of oil samples from the various vehicles selected (how many vehicles? how types of vehicles?) and, since there was a small number of samples collected (how many?). If only one sample (according to Figure 1) has been taken and the others are old, what is the point of the study? There is no continuity in sampling.

14. Line 186. This sentence: ‘it was felt the need to use data from older samples belonging to the same homogeneous group’. How do you ensure the old samples were collected under the same conditions as the new ones?

16. Figure 1 does not contribute anything. It should be converted into graphs that represent the variation of the useful data for the document.

17. Figures 1 and 2 are not adequately explained and developed in the text.

18. The columns shown in Figure 2 are not in chronological order (first 01/2008, then 07/2007, etc ...) Why ?.

19. Is Figure 2 necessary? The information the authors provide could be written in the text. The data is not used at all and also this figure is just an example of the data that is analyzed.

Author Response

Attached please find the responses.

Reviewer 2 Report

Reviewer believes that article needs some improvements for assure clarity and readability.

My main comments concern next points:

·        English grammar used on the manuscript requires some improvements. At times this makes it difficult to understand what the authors are trying to portray. Take care with specific jargon on the oil analysis topic.

·        Title proposed is quite confusing. Assess if that alternative could be acceptable for authors “Condition Monitoring with Prediction Based on Diesel engine oil analysis. A case study for urban buses”.

·        Paragraph on page 4, lines 154-156, must be clarified.

·        Parameters monitored on oil samples referred to ASTM or DIN standards are clear but which standards are PE-TA.XXX? Can you give more info about PQ Index (Adim)? If this parameter is not relevant, probably can be ignored. If it is relevant more info is required.

·        Data presented on table 2 and figure 3 are coming from the same vehicle/engine or are coming from different vehicles? Which is the reason for the iron concentration reduction between 22183 and 27682 km? Can be assumed that it has been a partial oil drain and refill or it is consequence just of oil topping up?

·        Authors stated (page 11, line 282) that: “The paper demonstrates that using condition monitoring maintenance, the intervals of the interventions can be increased….” But, which is the original oil drain interval for that engines/vehicles? And which is the oil drain enlargement obtained? That data has not been presented on the paper? Where is the demonstration stated?.

Author Response

Attached please find the responses.

Reviewer 3 Report

The abstract shall be a bit reworked in order to emphasise the modelling methods used - the mathematical tool.

The contents otherwise is interested due to the real data from in field operation.

Some relevant and topic related works shall be also mentioned:

Perspective analysis outcomes of selected tribodiagnostic data used as input for condition based maintenance. RELIABILITY ENGINEERING & SYSTEM SAFETY, 2016, vol. 145, no. 1, p. 231-242. ISSN 0951-8320.

Engine residual technical life estimation based on tribo data. Eksploatacja i Niezawodnosc – Maintenance and Reliability, 2014, vol. 16, no. 2, p. 203-210. ISSN 1507-2711.

Contribution to system failure occurrence prediction and to system remaining useful life estimation based on oil field data. Proceedings of the Institution of Mechanical Engineers, Part O: Journal of Risk and Reliability, 2015, vol. 229, no. 1, p. 36-45. ISSN 1748-006X.

The effect of soot and diesel contamination on wear and friction of engine oil pump. Tribol Int, 2017; 115: 285-296. 10.1016/j.triboint2017.05.041.

Lubricating oil conditioning sensors for online machine health monitoring – A review. Tribol Int, 2017; 109: 473-484. 10.1016/j.triboint.2017.01.015.

Author Response

Attached please find the responses.

Round 2

Reviewer 1 Report

The R1 version of the document has improved compared to the initial version. However, the document continues to have many deficiencies, but such shortcomings could be solved.

GENERAL

2. References. Minimize the use of documentation in Portuguese as references (there are eight references in Portuguese, 36% of the total is too much). Some references are somewhat old (1996, 1997, 1999, and 2000). The authors should look for more current references. In general, the number of references is scarce.

R: We added must references, according to the reviewer suggestion

Reviewer: This comment has been partially solved. Substituting bibliography in Portuguese is necessary.

5. Introduction. The introduction should include oil analysis methods, the influence of used fuel, and the objectives of the document.

R. The Introduction was changed according to the reviewer suggestion

Reviewer: This comment has not been solved

6. Figures. The graphics quality of the figures is very bad — figures without legends, with minuscule letter size, no name on the axes, etc. Please, improve these figures.

R: The figures and graphics was changed according to the reviewer suggestion

Reviewer: The Figures have not been improved, some of them continue being a screenshot, others as Figure 4, have a low quality.

SPECIFIC

1. Lines 33 to 35. Please move this sentence at the end of the introduction.

R: The lines 33 to 35 moved for the end of the introduction, was changed according to the reviewer suggestion

Reviewer: This sentence continues in the same place in the text. This comment has not been fixed.

3. Please, join these two sentences in one: ‘The condition monitoring maintenance appeared in the ’70-’80s to designating a new approach to planned maintenance based on the knowledge of the state of equipment, using condition monitoring techniques, [1]. Condition monitoring maintenance is the maintenance carried out using an evaluation of the equipment state, usually carried out continuously, [2].’ The information is repetitive and superfluous.

R: They were made some changes to clarify this question

Reviewer: This comment has not been solved

6. Currently, in fleet vehicles a certain percentage of biodiesel (or other alternative fuels) is used. It would be interesting for the authors to add articles to the introduction that analyse the effect of these fuels on oil. Some examples (of course the authors can look for other articles)

R: We added references, according to the reviewer suggestion

Reviewer: The commented articles have been included, but it was intended that the authors improve the introduction of the information of those articles. This comment has been partially solved.

7. Lines 118-121. Please add the studies that support this premise.

R: They were made some changes to clarify this question

Reviewer: This comment has not been solved

8. Lines 122-125. Please add the studies that support this premise.

R: They were made some changes to clarify this question

Reviewer: This comment has not been solved

10. Lines 140-142. Please add the studies that support this premise.

R: They were made some changes to clarify this question

Reviewer: This comment has not been solved

11. It would be interesting if the introduction of section 2 (Lines 116-142) were included in section 1.

R: They were made some changes to clarify this question

Reviewer: This comment has not been solved

13. Line 184. This sentence: ‘This monitoring was done through a periodic collection of oil samples from the various vehicles selected (how many vehicles? how types of vehicles?) and, since there was a small number of samples collected (how many?). If only one sample (according to Figure 1) has been taken and the others are old, what is the point of the study? There is no continuity in sampling.

R: We added information, according to the reviewer suggestion

Reviewer: This comment has been partially solved.

14. Line 186. This sentence: ‘it was felt the need to use data from older samples belonging to the same homogeneous group’. How do the authors ensure the old samples were collected under the same conditions as the new ones?

R: They were made some changes to clarify this question

Reviewer: This comment has not been solved

16. Figure 1 does not contribute anything. It should be converted into graphs that represent the variation of the user data for the document.

R: This is the original results document sent by the laboratory.

Reviewer: It is not necessary for the original document of the laboratory, the critical thing for the advance in the field is the interpretation of the results. Obviously, it is not necessary for me to show that the data is real, only the interpretation of the data is of interest to readers. The interpretation of the data is best made with graphs or tables.

17. Figures 1 and 2 are not adequately explained and developed in the text.

R: Was changed according to the reviewer suggestion

Reviewer: This comment has not been solved

NEW COMMENTS

Please, do not use the first-person plural (we) in the sentences. Lines 101, 165, 190, 249, 267.

The conclusions derived from the results are completely obvious.

Many of the suggested comments have not been resolved.

Author Response

Review 1

Summary of the changes

The R1 version of the document has improved compared to the initial version. However, the document continues to have many deficiencies, but such shortcomings could be solved.

GENERAL

2. References. Minimize the use of documentation in Portuguese as references (there are eight references in Portuguese, 36% of the total is too much). Some references are somewhat old (1996, 1997, 1999, and 2000). The authors should look for more current references. In general, the number of references is scarce.

  R: We added must references, according to the reviewer suggestion

 Reviewer: This comment has been partially solved. Substituting bibliography in Portuguese is necessary.

 R2: We added must references, according to the reviewer suggestion (references in Portuguese, 15%). More references and more recent / current have been introduced.

5. Introduction. The introduction should include oil analysis methods, the influence of used fuel, and the objectives of the document.

 R: The Introduction was changed according to the reviewer suggestion

 Reviewer: This comment has not been solved

 R2: The Introduction was changed according to the reviewer suggestion

“The paper starts with a global analysis about the importance of the oil analysis to predictive maintenance based on condition monitoring. Next, some examples of sheets of oil analysis, with emphasis of some important variables are presented.

Based on the preceding approach the paper follows the next steps:

·        First, the mathematical model to help predicting the next intervention based on Exponential Smoothing, the variable Fe is summarily presented as example;

·        Due to the variation of the variable Fe, a t-Student distribution with bilateral test of hypotheses is used;

·        Then an example using several values for smoothing parameter and some levels of significance is presented;

·        Finally, the influence of the maintenance policy, namely the predictive in the reserve fleet is discussed.”

6. Figures. The graphics quality of the figures is very bad — figures without legends, with minuscule letter size, no name on the axes, etc. Please, improve these figures.

 R: The figures and graphics was changed according to the reviewer suggestion

 Reviewer: The Figures have not been improved, some of them continue being a screenshot, others as Figure 4 has a low quality.

 R2: The figure/graphics 4 was changed according to the reviewer suggestion.

SPECIFIC

1. Lines 33 to 35. Please move this sentence at the end of the introduction.

 R: The lines 33 to 35 were moved for the end of the introduction, was changed according to the reviewer suggestion

 Reviewer: This sentence continues in the same place in the text. This comment has not been fixed.

 R2: We apologize, but would like to keep in the same place in the text.

3. Please, join these two sentences in one: ‘The condition monitoring maintenance appeared in the ’70-’80s to designating a new approach to planned maintenance based on the knowledge of the state of equipment, using condition monitoring techniques, [1]. Condition monitoring maintenance is the maintenance carried out using an evaluation of the equipment state, usually carried out continuously, [2].’ The information is repetitive and superfluous.

 R: They were made some changes to clarify this question

 Reviewer: This comment has not been solved

 R2: Were made some changes to clarify this question, the text has been improved.

“Condition monitoring maintenance is the maintenance carried out by using an evaluation of the equipment state, that appeared in the ’70-’80s to designate a new approach to planned maintenance based on condition monitoring techniques, [1-2].”

6. Currently, in fleet vehicles a certain percentage of biodiesel (or other alternative fuels) is used. It would be interesting for the authors to add articles to the introduction that analyse the effect of these fuels on oil. Some examples (of course the authors can look for other articles).

 R: We added references, according to the reviewer suggestion

 Reviewer: The commented articles have been included, but it was intended that the authors improve the introduction of the information of those articles. This comment has been partially solved.

 R2: Were made some changes to clarify this question, the text has been improved.

“Nowadays, in fleet vehicles, a certain percentage of biodiesel is used. It can be stated from several authors that the influence of this new type of fuel on the oil degradation is not consensual and, by consequence, on the maintenance based on condition monitoring. Some examples of papers that analyze the effect of these kind of fuels on oil are reported in [20-24].”

7. Lines 118-121. Please add the studies that support this premise.

 R: They were made some changes to clarify this question

 Reviewer: This comment has not been solved

 R2: Were made some changes to clarify this question

Regardless of the reference [33] is old and only published on paper, we consider that it contains a good case study to support their reference.

“A key feature of lubricants is their behavior with increasing temperature. They are not used at room temperature; the temperature and pressure are often high. The oils undergo a change when the temperature increases, and their degradation under operating conditions is a problem involving significant economic losses. To report certain special properties of the oil, or to improve the existing ones, especially when the lubricant is subjected to severe working conditions, chemicals are added (additives). The degradation of a lubricant is not an instantaneous process – the loss of its physicochemical properties and contamination are progressive over time and with the use of equipment along its lifetime. Lubricant degradation is affected by: oxidation; viscosity variation; contamination; loss of additives (anti-corrosion, anti-wear, dispersing agents, etc.) [33].”

 8. Lines 122-125. Please add the studies that support this premise.

 R: They were made some changes to clarify this question

.Reviewer: This comment has not been solved

 R2: They were made some changes to clarify this question

Regardless of the reference [33] is old and only published on paper, we consider that it contains a good case study to support their reference.

“Today’s high-performance lubricants do more than simply reduce friction and wear: they control the formation of deposits, control airborne contaminants, protect against corrosion, have a cleaning function, and maintain the proper operating temperature [33-40].”

10. Lines 140-142. Please add the studies that support this premise.

 R: They were made some changes to clarify this question

 Reviewer: This comment has not been solved

 R2: Were made some changes to clarify this question

The reference [41].

“             Lubricant analysis is regularly performed in some industries [41]. It involves four basic steps:

1.            Obtaining a sample

Collection of a representative sample of a lubricant, observing certain precautions such as: using clean and dry containers; taking extreme care during collection to prevent external contamination; taking samples at operating temperatures [41].”

11. It would be interesting if the introduction of section 2 (Lines 116-142) were included in section 1.

 Reviewer: This comment has not been solved

 R2: We apologize, but would like to keep in the same section.

13. Line 184. This sentence: ‘This monitoring was done through a periodic collection of oil samples from the various vehicles selected (how many vehicles? how types of vehicles?) and, since there was a small number of samples collected (how many?). If only one sample (according to Figure 1) has been taken and the others are old, what is the point of the study? There is no continuity in sampling.

 R: We added information, according to the reviewer suggestion

 Reviewer: This comment has been partially solved.

 R2: We added information, according to the reviewer suggestion, the text has been improved.

“This monitoring was done through periodic collection of oil samples from the various vehicles selected and, since there was a small number of samples collected during the period in which this monitoring was developed, the need to use data from older samples belonging to the same homogeneous group was felt. In this research, 10 standards (12 meters) urban passenger transport vehicles of three different brands were studied, having being analysed and studied 60 oil samples.

These samples helped to support the studies carried out and proved the relevance of the oil analysis to predictive maintenance based on oil condition monitoring as well as to support a new maintenance planning to be used by the company in the future.”

14. Line 186. This sentence: ‘it was felt the need to use data from older samples belonging to the same homogeneous group’. How do the authors ensure the old samples were collected under the same conditions as the new ones?

 R: They were made some changes to clarify this question

 Reviewer: This comment has not been solved

 R2: Were made some changes to clarify this question, the text has been improved.

“This monitoring was done through periodic collection of oil samples from the various vehicles selected and, since there was a small number of samples collected during the period in which this monitoring was developed, the need to use data from older samples belonging to the same homogeneous group was felt. In this research, 10 standards (12 meters) urban passenger transport vehicles of three different brands were studied, having being analysed and studied 60 oil samples.

These samples helped to support the studies carried out and proved the relevance of the oil analysis to predictive maintenance based on oil condition monitoring as well as to support a new maintenance planning to be used by the company in the future.”

16. Figure 1 does not contribute anything. It should be converted into graphs that represent the variation of the user data for the document.

 R: This is the original results document sent by the laboratory.

 Reviewer: It is not necessary for the original document of the laboratory, the critical thing for the advance in the field is the interpretation of the results. Obviously, it is not necessary for me to show that the data is real, only the interpretation of the data is of interest to readers. The interpretation of the data is best made with graphs or tables.

 R2: This is the original results document sent by the laboratory.

Considering that the others referees accepted the figure, we consider that, regardless of the pertinence of your comments we would like to maintain the figure

17. Figures 1 and 2 are not adequately explained and developed in the text.

 R: Was changed according to the reviewer suggestion

 Reviewer: This comment has not been solved

 R2: Were made some changes to clarify this question, the text has been improved.

“Subsequently, the reports of the results obtained from the various analyses carried out on the samples collected were received (Figure 1) - this is the original document with the results sent by the laboratory.

These analysis reports allow to control various properties of the lubricants and to evaluate their degradation throughout the life cycle of the equipment. From them, they can also follow the history of the analyses carried out over time. These include: antifreeze; appearance; fuel; content water; soot; nitration; oxidation; sulfation; viscosity; viscosity index; Total Base Number (TBN); wear metals (Al content, Cr, Fe, Mo, Na, Ni, Pb, Si, Sn, V); particles. Figure 1 also illustrates the history of the diverse variables studied, as, for example, the number of particles.”…

…“The data collected was entered into an Excel spreadsheet, in order to create a database where they could be analysed more easily.

Figure 2 shows an Excel database example, per vehicle. It can also be verified, in this figure, the historical data of the collected analyses referring to the equipment, as well as the identification and the characteristics of the vehicle studied.”

NEW COMMENTS

Please, do not use the first-person plural (we) in the sentences. Lines 101, 165, 190, 249, 267.

 R2: The text has been improved.

The conclusions derived from the results are completely obvious.

Many of the suggested comments have not been resolved.

Thanks for your extremely important contributions.

We read carefully your questions and contributions and we uploaded a paper that we hope answer to all of them.
